# Towards Omnidirectional Reasoning with 360-R1: A Dataset, Benchmark, and GRPO-based Method

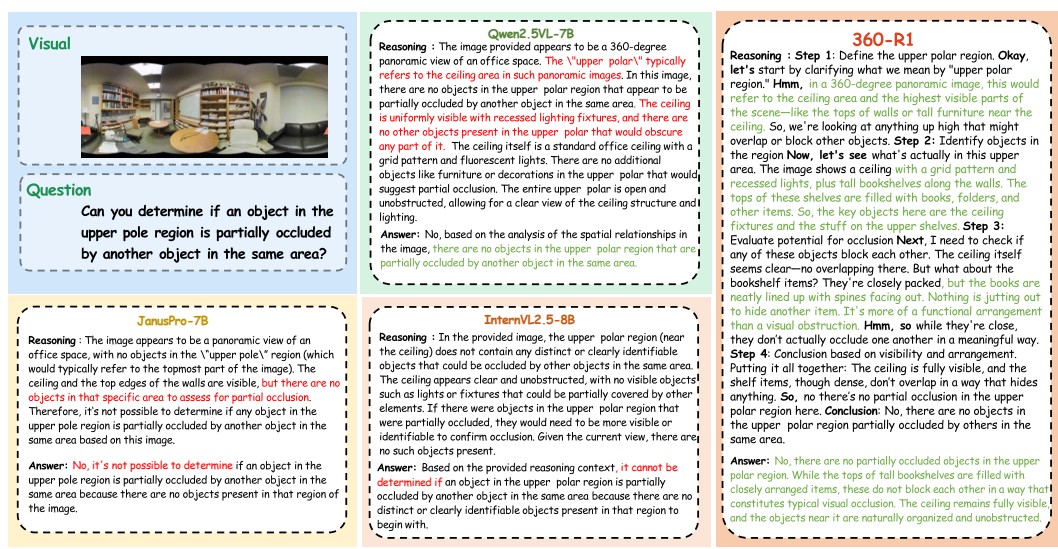

Figure 1: **Comparison of occlusion reasoning in 360° VQA.** Given a panoramic scene and a spatial question about whether an object in the upper pole region is partially occluded, four models provide different levels of reasoning. **360-R1** demonstrates the most precise and comprehensive reasoning, identifying relevant spatial elements and producing a correct answer.

## Abstract

Omnidirectional images (ODIs), with their 360° field of view, provide unparalleled spatial awareness for immersive applications like augmented reality and embodied AI. However, the capability of existing multi-modal large language models (MLLMs) to comprehend and reason about such panoramic scenes remains underexplored. This paper addresses this gap by introducing ***OmniVQA***, the ***first*** dataset and conducting the ***first*** benchmark for omnidirectional visual question answering. Our evaluation of state-of-the-art MLLMs reveals significant limitations in handling omnidirectional visual question answering, highlighting persistent challenges in object localization, feature extraction, and hallucination suppression within panoramic contexts. These results underscore the disconnect between current MLLM capabilities and the demands of omnidirectional visual understanding, which calls for dedicated architectural or training innovations tailored to 360 ° imagery. Building on the OmniVQA dataset and benchmark, we further introduce a reinforcement learning method with structured rewards, ***360-R1***, based on Qwen2.5-VL-Instruct. Concretely, we modify the group relative policy optimization (GRPO) by proposing three novel reward function, *i.e.*, reasoning process similarity reward, answer semantic accuracy reward, and format reward. Extensive experiments on our OmniVQA demonstrate the superiority of our proposed method in omnidirectional space. *OmniVQA will be open-sourced after review process; original images require Stanford 2D3D access. Code is in the supplement.*

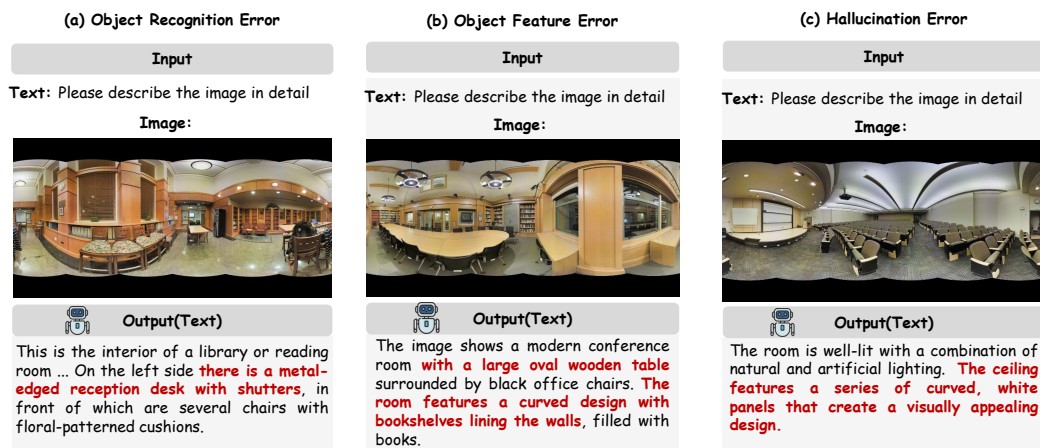

Figure 2: **Error Cases in Omnidirectional Captioning.** Three common errors by multimodal LLMs on 360° images: (a) misidentified objects; (b) incorrect object attributes or context; (c) hallucinated content unrelated to the image.

## 1 INTRODUCTION

Driven by the rapid advancement of Augmented Reality (AR), Virtual Reality (VR), and Embodied AI systems, coupled with a growing demand for immersive visual experiences, omnidirectional images (ODIs) have attracted significant research interest (Li et al., 2021a; Cao et al., 2024; Danieau et al., 2017; Li et al., 2021b; Su & Grauman, 2017; Coors et al., 2018). ODIs have been explored across a variety of downstream tasks, such as scene understanding (Zheng et al., 2023b;a; 2024; 2025; Zhang et al., 2024; Zhong et al., 2025). Unlike conventional 2D images with limited fields of view, ODIs capture the complete $180° \times 360°$ surroundings, encoding much richer spatial information, which makes them especially suitable for applications requiring holistic environmental understanding (Yun et al., 2021; Xiao et al., 2012).

Recently, significant progress has been made in developing strong and generalizable MLLMs (Han et al., 2024). These models are capable of integrating information from diverse modalities and have demonstrated impressive generalization across a wide range of downstream tasks (Ramesh et al., 2022; Gao et al., 2024; Yue et al., 2024a;b; Fang et al., 2025). For instance, the Visual Question Answering (VQA) serves as a crucial benchmark for learning usable real-world models and assessing models' ability in integrating visual comprehension of images with semantic understanding of questions, alongside the necessary reasoning capabilities (de Faria et al., 2023; Huynh et al., 2025). Despite these advances, VQA research remains constrained to conventional images, overlooking the unique advantages of omnidirectional vision, *i.e.*, the comprehensive understanding of the whole surrounding scenes (Antol et al., 2015; Hudson & Manning, 2019; Schwenk et al., 2022; Lu et al., 2022). Meanwhile, existing work (Chou et al., 2020) fails to benchmark the MLLMs' performance on panoramic domain.

In this paper, we propose a new omnidirectional visual question answering dataset, OmniVQA. The dataset focuses on key tasks such as object localization, object attribute analysis, and spatial relationship reasoning in polar regions. As illustrated in Figure 2, persistent challenges remain in accurately performing object recognition, attribute analysis, and spatial relationship description in panoramic views. These limitations lead to subpar performance by current MLLMs, as shown in Table 1, where JanusPro-7B achieves a relatively low DeepSeekScore. Furthermore, benchmarks tailored for omnidirectional VQA remain scarce, hindering evaluation and comparison in complex panoramic settings.

To construct the OmniVQA dataset, we leverage equirectangular projection (ERP) panoramic images from the 2D-3D-S dataset (Armeni et al., 2017). To address the geometric distortions inherent in panoramic imagery, we design three categories of questions: object identification, attribute analysis, and spatial relationship reasoning. Our data generation pipeline consists of three stages. First, a Qwen2.5-VL (Bai et al., 2025) model fine-tuned on OmniVQABench is used to generate detailed

visual descriptions. These descriptions are then passed to DeepSeek-R1 (DeepSeek-AI, 2025b) to produce Chain-of-Thought (CoT) (Wei et al., 2022) style reasoning. Finally, Qwen2.5-14B Instruct (Qwen Team, 2025) summarizes the reasoning into a concise answer. To ensure annotation quality, we adopt an iterative refinement strategy. Reasoning–answer pairs from fine-tuned and untuned models are compared using SentenceBERT-Score, and high-quality pairs (score > 0.8) are used to update the model. The process repeats, followed by manual correction of the remaining samples.

We also propose a post-training strategy based on reinforcement learning, termed 360-R1. Built upon the Qwen2.5-VL-Instruct model (Bai et al., 2025), which has already undergone extensive instruction tuning,360-R1 introduces structured reward functions specifically tailored for 360° visual comprehension. These include rewards for reasoning consistency, semantic answer accuracy, and output format compliance, all the semantic similarity automatically evaluated via the DeepSeek-V3 (DeepSeek-AI, 2025a) . We adopt the Group Relative Policy Optimization (GRPO) (Shao et al., 2024) algorithm to integrate these rewards while ensuring training stability. This strategy significantly enhances the model's ability to perform spatial reasoning, suppress hallucinations, and generate machine-readable responses in complex panoramic scenes 1. **(I)** *OmniVQA Dataset*. We introduce the first open-source omnidirectional VQA dataset built upon ERP-format panoramic images, containing three task types: object identification, attribute analysis, and spatial reasoning–particularly focused on polar regions. **(II)** *OmniVQABench*. A comprehensive benchmark designed to evaluate MLLMs in 360° visual environments. **(III)** *360-R1 Framework*. A reinforcement learning framework leveraging structured rewards and GRPO, significantly enhancing spatial reasoning and object identification.

## 2 RELATED WORK

**Omnidirectional Vision Question Answering (OVQA)** Visual Question Answering (VQA) aims to generate natural language answers given an image and a corresponding question (de Faria et al., 2023; Huynh et al., 2025). The widely used VQA v1.0 (Antol et al., 2015) includes both real-world and abstract scenes. VQA v2.0 (Goyal et al., 2017) mitigates language bias by balancing question distributions per image. To extend VQA to omnidirectional vision, the VQA 360° dataset (Chou et al., 2020) adapts Stanford 2D-3D (Armeni et al., 2017) and Matterport3D (Chang et al., 2017), using cube projection (CP) for panoramic representation. However, the dataset is not publicly released. In contrast, Pano-AVQA (Yun et al., 2021) targets 360° video-based QA, incorporating bounding-box-level annotations and enabling spherical spatial and audiovisual reasoning. Despite these efforts, publicly available, high-quality benchmarks for image-based omnidirectional VQA remain scarce, limiting thorough evaluation of models in panoramic scenarios. To address this gap, we introduce **OmniVQA**, the first open-source omnidirectional VQA dataset and benchmark, focusing on object identification, attribute analysis, and spatial reasoning in complex 360° environments.

**Multi-Modal Large Language Models** Multi-modal large language models (MLLMs) have shown strong performance in visual dialogue, image captioning, and visual question answering (Li et al., 2022; 2023; Caffagni et al., 2024; Das et al., 2016; Xu et al., 2015). InternVL 2.5 retains the ViT-MLP-LLM architecture while reducing visual tokens via pixel rearrangement, supporting multi-image and video input with techniques like dynamic high-resolution training and loss re-weighting (Chen et al., 2025b). LLaVA-OneVision (Li et al., 2024), the first open-source model effective across single-image, multi-image, and video tasks, combines Qwen-2 (Yang et al., 2024) and SigLIP (Zhai et al., 2023) using MLP, with a Higher AnyRes strategy for balanced performance and efficiency. Qwen2.5VL (Bai et al., 2025) introduces 2D-RoPE, window attention, dynamic FPS sampling, and MLP-based sequence compression for efficient video understanding. DeepSeek-VL2 (Wu et al., 2024b) employs a Mixture-of-Experts architecture with dynamic tiling and latent attention for high-resolution processing. Janus (Wu et al., 2024a) uses decoupled visual paths within a unified Transformer, while JanusPro (Chen et al., 2025a) extends this with prolonged pretraining and expanded multi-modal data for improved generation. In this paper, we propose the **360-R1** method, which incorporates reinforcement learning during post-training, specialized for 360° environments.

**Reasoning with CoT** Lu et al. (2022) pioneered few-shot CoT reasoning with manually annotated rationales. More recently, reasoning-oriented large language models (RLLMs) like OpenAI o1 (OpenAI, 2024b) and DeepSeek R1 (DeepSeek-AI, 2025b) have driven extensive research on long CoT reasoning. In the vision-language domain, Vision-R1 (Huang et al., 2025) introduces multimodal

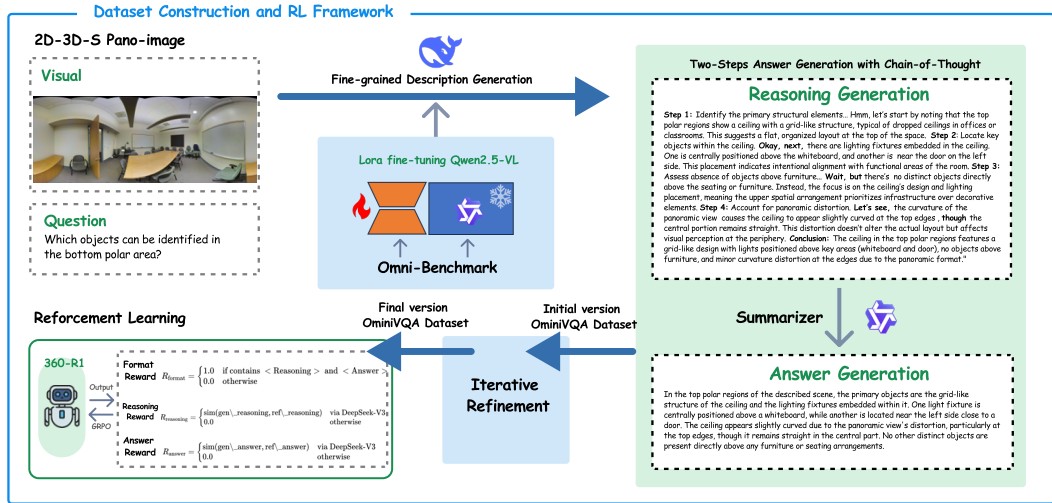

Figure 3: Overview of the OmniVQA Dataset Construction and 360-R1 Framework.

CoT by converting visual inputs into structured reasoning chains, while R1-OneVision (Yang et al., 2025) proposes a cross-modal CoT pipeline for step-by-step reasoning. LLaVA-CoT (Xu et al., 2024) uses a multi-stage approach with stage-level beam search for complex multimodal tasks, and LlamaV-o1 (Thawakar et al., 2025) incorporates curriculum learning for efficient multi-step reasoning. In this paper, our **OmniVQA** dataset provides two-step CoT annotations–reasoning and answer summaries–to guide the reinforcement learning of the **360-R1** framework.

## 3 OMNIVQA: DATASET AND BENCHMARK

### 3.1 OVERVIEW

The OmniVQA dataset is the first open-source dataset in the field of omnidirectional visual question answering , introducing novel challenges such as object attribute identification and spatio-temporal reasoning. It extends traditional VQA tasks by incorporating panoramic scene understanding and advanced spatial reasoning, enabling multi-modal interpretation of complex, immersive environments. Developed based on the Stanford 2D-3D-S dataset (Armeni et al., 2017), which covers over 6,000 m² of indoor scenes, OmniVQA dataset contains 1,213 panoramic images in equirectangular projection (ERP) format at a resolution of 1440×720, preserving angular fidelity and geometric structure. A total of 4,852 VQA pairs are included, distributed as follows:31.12% (1,510 ) questions focus on object identification, 25.87% (1,255) analyze object attributes, and 43.01% (2,087) probe spatial relationship reasoning.

### 3.2 DATASET CONSTRUCTION

**Question Design.** The OmniVQA dataset is constructed based on panoramic images from the 2D-3D-S dataset (Armeni et al., 2017). To address the three typical error types illustrated in Figure 2, we design three corresponding types of questions. The first type focuses on object identification, which requires recognizing objects within the panoramic scene, particularly those located in the polar regions where distortions are most prominent. The second type involves object attribute analysis, which asks the model to describe visual characteristics such as shape, size, and material. The third type emphasizes spatial relationship reasoning, where the goal is to infer spatial arrangements, occlusion states, and interactions between objects. Table 7 lists these categories along with their distribution percentages.

Compared with outdoor VQA datasets that typically contain sparse visual content (Mirowski et al., 2019), the OmniVQA dataset emphasizes dense indoor scenes. In these environments, the distortions introduced by equirectangular projection in the polar areas significantly hinder accurate visual

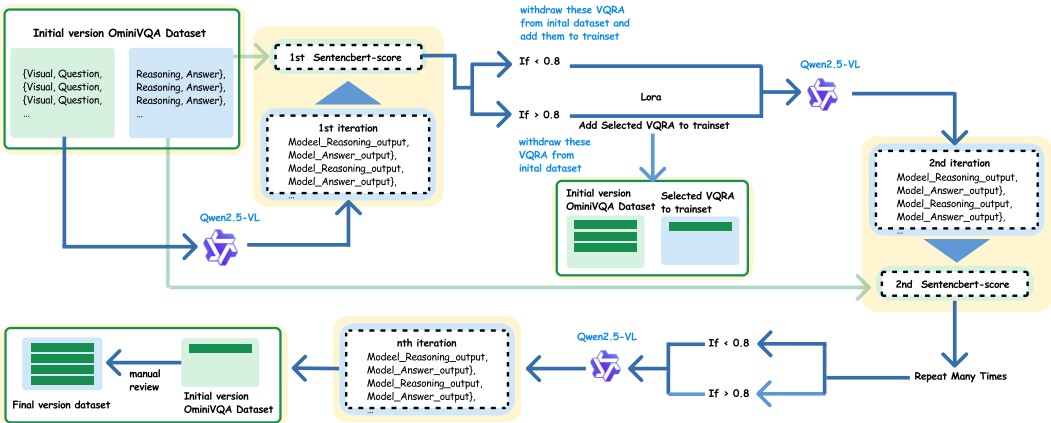

Figure 4: Iterative Refinement Pipeline.

perception. Our question design specifically addresses these challenges and aims to enhance the model's ability to understand panoramic scenes under geometric distortions. A detailed breakdown of question types is presented in appendix.

**Two Steps Answer Generation with Chain-of-Thought (Wei et al., 2022).** To improve the quality of the answer in the presence of distortions in the panoramic image, we adopt a two-step reasoning pipeline, as illustrated in Figure3. In the first step, a Qwen2.5-VL-7B model is fine-tuned using low-rank adaptation (LoRA) (Hu et al., 2021) on our OmniVQABench dataset to generate detailed visual descriptions of each panoramic image. In the second step, the DeepSeek-R1 model takes these descriptions as input and generates structured CoT reasoning sequences that capture spatial and semantic relationships. Finally, the Qwen2.5-14B model summarizes CoT reasoning into concise answers. This modular pipeline enhances robustness by structuring the reasoning process into distinct perception and reasoning stages, which improves accuracy under panoramic distortion.

**Iterative Refinement.** To further improve dataset quality, we we introduce an iterative refinement loop, as shown in Figure 4. First, we compare reasoning-answer pairs generated by two models: the fine-tuned Qwen2.5-VL-7B (trained on OmniVQABench) and the original Qwen2.5-VL-7B (without fine-tuning). Both outputs are summarized into final answers using Qwen2.5-14B. Then, we evaluate the semantic similarity between the two outputs using SentenceBERT-Score (Reimers & Gurevych, 2019), applied separately to the reasoning and the answer components. These scores are fused using an F1-style aggregation metric. Pairs with scores above 0.8 are selected to fine-tune a new model instance, initialized from scratch. This updated model is used to generate outputs for the remaining questions, and the evaluation process is repeated. High-confidence pairs are incrementally added to the training set. The cycle continues until only a small set of low-confidence pairs remains, which are manually verified and corrected to ensure overall consistency and accuracy.

# 4 MULTI-MODAL OMNIDIRECTIONAL VQA BENCHMARK

To construct a reliable benchmark for multi-modal omnidirectional VQA, we employ the QwenVL model to generate answers for panoramic images and their corresponding sampled questions. By analyzing these outputs, we identify three major error types, as illustrated in Figure 2. Guided by the distribution of these error types, we select relevant images from the Stanford 2D-3D-S dataset (Armeni et al., 2017), maintaining a 2:1:1 ratio across the three error categories. A few error-free samples are also included to ensure diversity, resulting in a benchmark of 200 panoramic images. To generate fine-grained visual descriptions for the 200 selected images, GPT-4o (OpenAI, 2024a) is prompted using several high-quality examples.

All descriptions are manually verified, with particular focus on correcting panoramic distortions and restoring object-level details. The DeepSeek-R1 model takes these descriptions as input and generates structured CoT reasoning. Based on those structured CoT-style reasoning, we then use Qwen2.5-14B to generate concise answers. DeepSeek-V3 is subsequently employed to review the

Table 1: Performance of SOTA MLLMs and our method on the OmniVQA Benchmark. The evaluation is split into **Open-ended QA** (measured by SBERTScore and DeepSeekScore) and **MCQ Accuracy**. Best results are in **bold**, second best are underlined.

| Model | Open-ended QA | | | | | | MCQ Acc. (%)↑ |
|---|---|---|---|---|---|---|---|
| | SBERTScore↑ | | | DeepSeekScore↑ | | | |
| | R | A | F1 | R | A | F1 | |
| **Close-source Model** | | | | | | | |
| GPT-4o | 0.5952 | 0.6569 | 0.6245 | **0.4971** | 0.5653 | 0.5290 | 63.33 |
| **Open-source Models** | | | | | | | |
| *LlavaOnevision Series (Li et al., 2024)* | | | | | | | |
| LlavaOnevision Qwen2-7b-ov | 0.4973 | 0.5802 | 0.5355 | 0.2336 | 0.2509 | 0.2419 | 31.11 |
| LlavaOnevision Qwen2-7b-si | 0.4758 | 0.5661 | 0.5170 | 0.1714 | 0.2533 | 0.2044 | 30.00 |
| *Janus Series (Wu et al., 2024a; Chen et al., 2025a)* | | | | | | | |
| JanusPro-7B | 0.5401 | 0.6379 | 0.5849 | 0.3262 | 0.3614 | 0.3429 | 62.22 |
| JanusPro-1B | 0.4079 | 0.5122 | 0.4541 | 0.1900 | 0.3240 | 0.2395 | 18.89 |
| Janus-1.3B | 0.4716 | 0.6007 | 0.5284 | 0.2119 | 0.2468 | 0.2281 | 22.22 |
| *InternVL Series (Chen et al., 2025b)* | | | | | | | |
| InternVL2.5-8B | 0.4932 | 0.6218 | 0.5500 | 0.3124 | 0.3403 | 0.3363 | 66.67 |
| InternVL2.5-4B | 0.5124 | 0.6230 | 0.5623 | 0.2289 | 0.2972 | 0.3029 | 54.44 |
| InternVL2.5-2B | 0.4822 | 0.6150 | 0.5406 | 0.2961 | 0.3246 | 0.3203 | 14.44 |
| *DeepSeek Series (Lu et al., 2024; Wu et al., 2024b)* | | | | | | | |
| DeepSeekVL2-small | 0.4707 | 0.6252 | 0.5370 | 0.4102 | 0.4554 | 0.4316 | 58.89 |
| DeepSeekVL-3b-chat | 0.4312 | 0.5100 | 0.4673 | 0.2164 | 0.2889 | 0.2476 | 45.56 |
| DeepSeekVL-1.3b-chat | 0.2866 | 0.4246 | 0.3422 | 0.1050 | 0.1255 | 0.1143 | 24.44 |
| *Qwen Series (Wang et al., 2024; Bai et al., 2025)* | | | | | | | |
| QwenVL2.5-7B-Instruct | 0.5870 | 0.6687 | 0.6252 | 0.4638 | 0.5555 | 0.5055 | 67.78 |
| QwenVL2.5-3B-Instruct | 0.5083 | 0.6549 | 0.5724 | 0.3625 | 0.4008 | 0.3966 | 56.67 |
| QwenVL2-7B-Instruct | 0.4812 | 0.6341 | 0.5472 | 0.3240 | 0.3711 | 0.3675 | 54.44 |
| QwenVL2-2B-Instruct | 0.4800 | 0.6373 | 0.5476 | 0.3067 | 0.3456 | 0.3461 | 56.67 |
| **360-R1 (Ours)** | **0.6100** | **0.6705** | **0.6388** | 0.4847 | **0.6282** | **0.5472** | **68.89** |

reasoning-answer pairs, ensuring logical consistency and semantic alignment. Finally, to avoid the limitations of evaluating model capabilities on a single question format, we randomly selected 90 of these samples and converted them into a multiple-choice question (MCQ) set for supplementary evaluation.

# 5 360-R1 FRAMEWORK

To enhance multi-modal reasoning capabilities in omnidirectional environments, we propose a reinforcement learning post-training strategy with structured rewards. Given that Qwen2.5-VL-Instruct (Bai et al., 2025) has already undergone extensive instruction tuning on large-scale datasets–ensuring stable, compliant generation–we directly apply RL on this model to promote structured reasoning and enforce output validity.

## 5.1 REWARD FUNCTION DESIGN

We design three reward functions to guide reinforcement learning using Group Relative Policy Optimization. These rewards aim to improve the quality of reasoning, the accuracy of answers, and the consistency of output formatting, thus enhancing the model's generalization and robustness.

**Reasoning Process Similarity Reward** evaluates the semantic and logical alignment between the generated reasoning and a reference COT reasoning. It extracts reasoning segments enclosed within predefined tags and computes a similarity score using the DeepSeek-V3 (DeepSeek-AI, 2025a) model via prompt-based evaluation. The score, ranging from 0.0 to 1.0, is tolerant to surface-level

variations while sensitive to logical discrepancies, providing fine-grained feedback to guide reasoning generation.

**Answer Semantic Accuracy Reward** measures the semantic similarity between the generated and reference answers. Answer content is automatically extracted from the structured output, and a similarity score is computed using DeepSeek-V3 (DeepSeek-AI, 2025a). The reward encourages the generation of semantically correct responses, even when lexical or syntactic differences exist.

**Format Reward.** This binary reward (1.0 or 0.0) checks whether the output adheres to a predefined structured format. It verifies the presence, order, and correct nesting of essential components such as reasoning and answer sections.

## 5.2 GROUP RELATIVE POLICY OPTIMIZATION

We adopt Group Relative Policy Optimization (GRPO) to achieve stable policy updates while effectively incorporating structured rewards. Unlike standard Proximal Policy Optimization (PPO), GRPO eliminates the need for a separate value function by estimating the advantage through group-wise normalized rewards. For each input question $q$, GRPO samples a group of $G$ responses $\{o_1, \ldots, o_G\}$ from the old policy $\pi_{\text{old}}$. A reward model assigns scalar scores $\{r_1, \ldots, r_G\}$, which are then normalized within the group:

$$\hat{r}_i = \frac{r_i - \text{mean}(r)}{\text{std}(r)}, \tag{1}$$

and used as the advantage for all tokens in the $i$-th response:

$$\hat{A}_{i,t} = \hat{r}_i. \tag{2}$$

The clipped surrogate loss is computed as:

$$\mathcal{L}_{\text{clip}} = -\mathbb{E}\left[\min\left(r_t \cdot \hat{A}_t, \text{clip}(r_t, 1 - \epsilon, 1 + \epsilon) \cdot \hat{A}_t\right)\right], \tag{3}$$

where $r_t = \frac{\pi_\theta(o_t|q, o_{<t})}{\pi_{\text{old}}(o_t|q, o_{<t})}$ is the token-level probability ratio. To prevent the updated policy from deviating too far from the reference policy, a KL divergence regularization term is included:

$$\mathcal{L}_{\text{GRPO}}(\theta) = \mathcal{L}_{\text{clip}} + \beta \cdot \text{KL}(\pi_\theta \| \pi_{\text{ref}}). \tag{4}$$

This group-relative formulation aligns well with reward models trained on preference data, and avoids the computational burden of training a separate value network. The combination of group-based advantage, clipping, and KL regularization ensures stable and efficient policy updates under supervision.

## 6 EXPERIMENT

**Experimental Setup.** We conduct our experiments on the OmniVQABench for evaluation, using the OmniVQA dataset for training. Our primary model, 360-R1, is built upon the Qwen2.5-VL-7B-Instruct base model. We benchmark our method against a comprehensive suite of state-of-the-art vision-language models, including the Qwen Series (QwenVL2-2B to QwenVL2.5-7B) (Wang et al., 2024; Bai et al., 2025), Llava-Onevision Series (Qwen2-7B-OV, Qwen2-7B-SI) (Li et al., 2024), Janus Series (1B to 7B) (Wu et al., 2024a; Chen et al., 2025a), InternVL2.5 Series (2B to 8B) (Chen et al., 2025b), and DeepSeekVL Series (1.3B to 7B) (Lu et al., 2024; Wu et al., 2024b).

**Implementation Details.** Our training strategy employs reinforcement learning using Group Relative Policy Optimization (GRPO) with lora. The training is conducted with a learning rate of 1e-5 and utilizes bf16 mixed precision. Crucially, the vision encoder remains frozen throughout the training process to preserve its powerful pre-trained features. The entire GRPO training runs for 550 steps (approximately 1 epoch) on 4×H800 GPUs, taking about 2.8 days. Full hyperparameter settings and training scripts are provided in the supplementary material.

**Evaluation Metrics.** Performance is evaluated based on the question format. For open-ended Visual Question Answering (VQA), we report **SentenceBERTScore** (Reimers & Gurevych, 2019) and **DeepSeekScore**. SentenceBERTScore measures sentence-level semantic similarity, while DeepSeekScore utilizes the `DeepSeekV3-chat` model (DeepSeek-AI, 2025a) to assess both the quality of the reasoning process and the accuracy of the final answer, outputting a score between

Table 2: Performance comparison between our 360-R1 and the QwenVL2.5-7B-Instruct baseline.

| Model | Open-ended QA | | | | | | MCQ Acc. (%) ↑ |
| | SBERTScore↑ | | | DeepSeekScore↑ | | | |
| | R | A | F1 | R | A | F1 | |
|---|---|---|---|---|---|---|---|
| QwenVL2.5-7B | 0.5870 | 0.6687 | 0.6252 | 0.4638 | 0.5555 | 0.5055 | 67.78 |
| **360-R1 (Ours)** | **0.6100** | **0.6705** | **0.6388** | **0.4847** | **0.6282** | **0.5472** | **68.89** |
| **Improvement** | **+3.92%** | **+0.27%** | **+2.18%** | **+4.51%** | **+13.09%** | **+8.25%** | **+1.11pp** |

0.0 and 1.0. For this LLM-based metric, we report the individual reasoning (R) and answer (A) scores, as well as their harmonic mean (F1), to provide a comprehensive evaluation. For multiple-choice questions (MCQ), we report standard Accuracy.

## 6.1 RESULTS ANALYSIS

Our model, 360-R1, establishes a new state-of-the-art for open-source models on the OmniVQA benchmark, demonstrating significant improvements over the strong QwenVL2.5-7B-Instruct baseline and outperforming all other open-source competitors across both open-ended and multiple-choice tasks. The detailed results are presented in Table 1.

**Open-ended QA Performance.** In the open-ended QA evaluation, 360-R1 shows marked superiority. For the **DeepSeekScore**, our model achieves the top F1 score of **0.5472**, representing a substantial 8.25% relative improvement over the baseline's 0.5055. This gain is driven by a remarkable +13.09% increase in the answer quality sub-score (A), where our model scores **0.6282**, surpassing even the powerful closed-source GPT-4o. The reasoning score (R) also improves by 4.51% relative to the baseline, reaching **0.4847**, second only to GPT-4o. For the **SBERTScore**, 360-R1 achieves the highest F1 score of **0.6388** among all models, indicating superior semantic fluency and alignment.

**Multiple-Choice Question (MCQ) Performance.** In the MCQ task, 360-R1 again leads the field, achieving the highest accuracy of **68.89%**. This result surpasses the strong baseline by 1.11 percentage points and outperforms other competitive models like InternVL2.5-8B (66.67%), confirming its robust understanding in a constrained-choice setting.

**Overall Effectiveness.** We attribute these comprehensive improvements to two key factors: the use of GRPO and the OmniVQA dataset. GRPO enables the model to learn from relative quality comparisons, effectively optimizing for both reasoning soundness and answer precision. The OmniVQA dataset, with its diverse and complex reasoning queries, provides the challenging training ground necessary to build these advanced capabilities. The results validate that our training approach significantly enhances both explicit reasoning quality and implicit semantic fidelity. Notably, our 7B parameter model not only sets a new benchmark for open-source models but also proves to be highly competitive with, and in several key aspects superior to, the much larger GPT-4o model.

## 6.2 ABLATION STUDY

To understand the effects of model initialization and reward design, we conduct ablation studies on: (1) comparing base models with different parameter scales, and (2) analyzing the impact of reward weight configurations. We also analyze the contribution of our training strategy and the model's cross-dataset generalization. For fair comparison, models in the first two studies are fine-tuned using GRPO for 200 steps.

**Impact of Model Parameter Scale.** As shown in Table 5, we compare two variants: 360-R1-3B and 360-R1-7B, based on QwenVL2.5-3B and QwenVL2.5-7B respectively. The larger 7B model consistently outperforms its 3B counterpart. Specifically, SentenceBERTScore-F1 improves by 7.69%, and DeepSeekScore-F1 shows a substantial gain of 18.80%. This demonstrates the significant benefits of scaling in pre-trained vision-language models, particularly for enhancing reasoning capabilities as measured by LLM-based metrics.

**Impact of Reward Weight Configuration.** We investigate how different reward weightings (format: Base:Reasoning:Answer) affect performance (Table 6). Compared to our default balanced setting (0.1:0.45:0.45), emphasizing the answer (0.1:0.4:0.5) slightly improves SentenceBERTScore-F1 (+0.82%) but degrades DeepSeekScore-F1 (-2.10%), suggesting a trade-off where semantic fluency is gained at the cost of reasoning quality. Conversely, prioritizing reasoning (0.1:0.5:0.4) increases DeepSeekScore-F1 (+1.29%) but lowers SentenceBERTScore-F1. This highlights the framework's sensitivity to reward design and validates our balanced configuration, which offers the most stable and competitive performance.

**Analysis of Training Strategy.** To isolate the contribution of our reinforcement learning stage, we compare our final 360-R1 model against a strong baseline trained only with Supervised Fine-Tuning (SFT) on the OmniVQA dataset. As detailed in Table 3, the GRPO-trained 360-R1 consistently outperforms the SFT-only model. The DeepSeekScore-F1 increases from 0.5085 to 0.5472 (+7.61%), and the SentenceBERTScore-F1 improves from 0.6264 to 0.6388 (+1.98%). These results clearly demonstrate that while SFT provides a solid foundation, GRPO is crucial for refining the model's reasoning process and enhancing the quality of its answers.

**Analysis of Cross-Dataset Generalization.** To assess generalization, we trained a model exclusively on the CFPano dataset (Zhang et al., 2025) and evaluated it directly on our `OmniVQABench`. As shown in Table 4, the model demonstrates strong generalization, achieving a SentenceBERT-F1 of 0.6252 and a DeepSeekScore-F1 of 0.5099. This performance surpasses several strong baselines from Table 1. However, it does not match our final in-domain-tuned 360-R1 model, suggesting that while our methodology imparts robust, generalizable knowledge, in-domain fine-tuning remains crucial for achieving state-of-the-art results.

Table 3: Comparison of training methodologies: SFT vs. GRPO (RL).

| Evaluation Metric (F1 Score) | SFT | 360-R1 (GRPO) |
|---|---|---|
| SBERTScore | 0.6264 | 0.6388 |
| DeepSeekScore | 0.5085 | 0.5472 |

Table 4: Cross-dataset generalization. Model trained on `CFPano` and evaluated on `OmniVQABench`.

| Evaluation Metric (F1 Score) | R | A | F1 |
|---|---|---|---|
| SBERTScore | 0.5947 | 0.6590 | 0.6252 |
| DeepSeekScore | 0.4800 | 0.5438 | 0.5099 |

Table 5: Comparison of Model Parameters (trained for 200 steps).

| Evaluation Metric(F1 Score) | 360-R1-3B | 360-R1-7B |
|---|---|---|
| SBERTScore | 0.5872 | 0.6324 |
| DeepSeekScore | 0.4292 | 0.5099 |

Table 6: Comparison of Different Reward Weight Configurations (360-R1-7B, trained for 200 steps).

| Evaluation Metric(F1 Score) | Format:Reasoning:Answer | | |
|---|---|---|---|
| | 0.1:0.45:0.45 | 0.1:0.4:0.5 | 0.1:0.5:0.4 |
| SBERTScore | 0.6324 | 0.6376 | 0.6261 |
| DeepSeekScore | 0.5099 | 0.4992 | 0.5165 |

# 7 CONCLUSION AND FUTURE WORK

In this paper, we introduced OmniVQA, the first VQA dataset specifically designed for 360° panoramic images, alongside its corresponding benchmark, OmniVQABench. Our proposed 360-R1 method, which leverages reinforcement learning with GRPO, significantly improves reasoning accuracy and semantic consistency in omnidirectional contexts, with ablation studies validating our structured reward functions.

Our *future research* will directly address the limitations of the current work. To overcome the dataset's limited scale ( 4.8K pairs) and its focus on indoor environments, we plan to expand OmniVQA to include diverse outdoor and dynamic scenes, thereby enhancing model generalization. Similarly, to counter the high computational cost of the GRPO framework, we will focus on developing more adaptive and computationally efficient reward mechanisms and optimization strategies. Finally, we aim to further enrich omnidirectional reasoning capabilities by integrating additional modalities, such as audio cues and temporal information, paving the way for more sophisticated real-world applications.

ETHICS STATEMENT

All authors have read and adhered to the ICLR Code of Ethics. Our research utilizes the 2D-3D-S dataset, which consists of panoramic images of indoor environments. While this dataset is intended for research and does not primarily feature individuals, we acknowledge the inherent privacy considerations when working with images of real-world spaces. The models developed and used in this work are built upon large-scale pre-trained models, and we recognize that such models can perpetuate societal biases present in their training data. Our benchmark, OmniVQA, is currently limited to indoor scenes from a specific dataset, which may not represent global diversity. We encourage future work to evaluate and mitigate potential biases when applying these models to broader contexts. The capabilities for detailed scene understanding could be misused for surveillance, and we release our work with the intention of advancing research in assistive technologies and immersive AI, advocating for the responsible and ethical application of these technologies.

REPRODUCIBILITY STATEMENT

To ensure the reproducibility of our results, we have made our best efforts to provide all necessary components. **Code:** The code for our 360-R1 framework, including the GRPO training scripts and evaluation logic, is included in the supplementary material and will be released in a public repository after review process. **Data:** The question-answer pairs of our OmniVQA dataset and the OmniVQABench will be made publicly available. As stated in the abstract, the original panoramic images are from the Stanford 2D-3D-S dataset, and access must be obtained directly from the original creators. The detailed data generation pipeline, including the iterative refinement process, is described in Section 3.2 and illustrated in Figure 4. **Models and Environment:** Our 360-R1 model is fine-tuned from the publicly available Qwen2.5-VL-7B-Instruct model. All models used for benchmarking are clearly cited in Section 6. Key hyperparameters, software dependencies, and hardware specifications are detailed in the supplementary material to facilitate the replication of our training environment. **Evaluation:** The evaluation metrics, including the specific models used for LLM-based scores (e.g., DeepSeek-V3), are described in Section 6. The full benchmark, including the multiple-choice question set, will be released to allow for standardized comparisons.

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

# A   PROMPT TEMPLATES

This part collects all of the system and user prompts we use to drive our VQA pipeline. Each prompt is encapsulated in a styled box for clarity, and includes guidance on inputs, expected outputs, and formatting conventions.

## A.1   FINE-GRAINED DESCRIPTION GENERATION (PERCEPTION STAGE)

**Role in Experiment:** This prompt is used to elicit a step-by-step, fine-grained analytical description of a panoramic image from a LoRA-fine-tuned `Qwen2.5VL-7B-Instruct` model. It provides the initial perception stage input for subsequent reasoning and answer generation modules. Detailed spatial relationship language is required to mitigate distortion-related errors and support downstream Chain-of-Thought (CoT) reasoning.

---

**Input:** Image and natural-language question

```
Please generate a fine-grained analytical description based on the
    image
and question provided below. Clearly present the complete thought
    process
rather than merely providing a final answer. When the question
    involves
spatial relationships, explicitly describe them using clear
    directional
terms such as 'up', 'down', 'left', 'right', 'front', 'back', '
    covering',
or 'adjacent to'.
```

---

## A.2   CHAIN-OF-THOUGHT (COT) REFORMATTING

**Role in Experiment:** This prompt is applied to convert free-form descriptive reasoning into a structured, multi-step Chain of Thought using the `DeepSeek-R1` model. It enforces explicit logical progression, which facilitates downstream evaluation, answer summarization, and reasoning-based reward functions.

```
System:

You are a Chain of Thought (CoT) reformatting expert. Your task is
    to
transform descriptive reasoning into natural, step-by-step thinking
that reflects how one would logically work through a problem. Ensure
each step builds on the previous one, retains all critical
    information,
and concludes with a direct answer to the original question.

User:

Guidelines:
1. Structure content into 3-5 explicit reasoning steps.
2. Use transition words (e.g., "First," "Next," "Then") and thinking
    -style
  phrases (e.g., "Hmm," "Let's see," "Okay").
3. Ensure logical flow: each step derives from the prior one.
4. Preserve all original information.
5. End with a clear conclusion.

Question: {question}
Original reasoning: {reasoning}

Transformed COT reasoning:
```

## A.3  CoT Summarisation (Answer Synthesis)

**Role in Experiment:** This prompt guides the `Qwen2.5-14B-Instruct` model to convert a structured Chain of Thought into a concise, factual answer. It ensures that the final answer is directly grounded in the provided reasoning context, preventing hallucination or unsupported information.

```
{"role":"system","content":"You are a precision-focused assistant.
    Generate concise
and factual answers based on the given question and reasoning
    context.
Answer should directly address the question using ONLY information
    from the reasoning."},
{"role":"user","content":prompt}
```

## A.4  Answer Consistency Review (VQA Evaluation)

**Role in Experiment:** This prompt uses `DeepSeek-V3` to evaluate whether a given answer is consistent with the original question and structured reasoning. It is critical for post-hoc filtering, reward calculation during RL post-training, and ensuring dataset quality.

```
{"role":"system","content":"You are an expert reviewer in the Visual
    Question
Answering (VQA) domain. Your task is to evaluate whether the
    provided answer
is consistent with the question and the structured reasoning.

Evaluation criteria:
1. Factual Accuracy
2. Logical Consistency
3. Completeness

If errors exist, provide a corrected version. If fully consistent,
    reply only:
CONSISTENT."},
{"role":"user","content":"Question: {question}
Reasoning: {reasoning}
Current Answer: {answer}

Please review item #{index}.
If not consistent, provide corrected answer. If consistent, reply:
    CONSISTENT."}
```

## A.5  SEMANTIC CONSISTENCY SCORING

**Role in Experiment:** This prompt queries `DeepSeek-V3` to compute a fine-grained semantic similarity score (range: 0.0000–1.0000) between reference and candidate answers. This score is used in both dataset filtering (e.g., during iterative refinement) and as a reward signal in RL training.

```
{"role":"system","content":"You are a professional evaluation
    assistant.
Analyze the semantic consistency between reference and candidate
    answers.
Guidelines:
1. Score range: 0.0-1.0
2. Consider accuracy and context
3. Return ONLY the numeric score with 4 decimal places
4. No additional text."},
{"role":"user","content":"Reference: {reference}\nCandidate: {
    candidate}\nScore:"}
```

## A.6  REASONING SIMILARITY SCORING (GRPO REWARD)

**Role in Experiment:** This prompt is used during Group Relative Policy Optimization (GRPO) training to compute the semantic similarity between the generated and reference reasoning chains, using `DeepSeek-Chat`. The returned score is directly used as the Reasoning Process Similarity Reward.

**System:**

```
You are an AI assistant that evaluates the similarity between two
    reasoning processes. Your response MUST follow the format '
    Similarity score: <score>', where <score> is a single floating-
    point number between 0.0 and 1.0 representing the similarity
    score.
```

**User:**

```
Compare these two reasoning passages and return a similarity score
    0-1.

Generated:
{gen_text}

Reference:
{ref_text}
```

## A.7 ANSWER SIMILARITY SCORING (GRPO REWARD)

**Role in Experiment:** This prompt is used during GRPO training to assess the semantic similarity between the generated answer and the reference answer, using `DeepSeek-Chat`. The score forms the Answer Semantic Accuracy Reward.

**System:**

```
You are an AI assistant that evaluates the similarity between two
    answer. Your response MUST follow the format 'Similarity score:
    <score>', where <score> is a single floating-point number
    between 0.0 and 1.0 representing the similarity score.
```

**User:**

```
Compare these two answer texts and return a similarity score 0-1.

Generated:
{gen_text}

Reference:
{ref_text}
```

## A.8 FINE-GRAINED DESCRIPTION PROMPT FOR GPT-4O (FEW-SHOT)

**Role in Experiment:** This prompt is used to guide GPT-4o in generating high-quality, fine-grained descriptions of 360-degree panoramic images. The prompt provides multiple few-shot examples, each consisting of a panoramic image and a detailed description that explicitly addresses distortion effects and spatial relationships.

**System:**

You are an expert visual-language model that generates fine-grained
    descriptions of 360-degree panoramic indoor scenes. Your goal is
     to identify furniture, materials, lighting, and spatial layout
    in natural and coherent English.

**User (Few-shot):**

Please generate a paragraph that accurately describes the scene
    shown in the image. Follow the style of the examples provided,
    focusing on object identities, spatial relationships, and the
    contextual organization of the scene. Be as specific and
    spatially precise as possible.

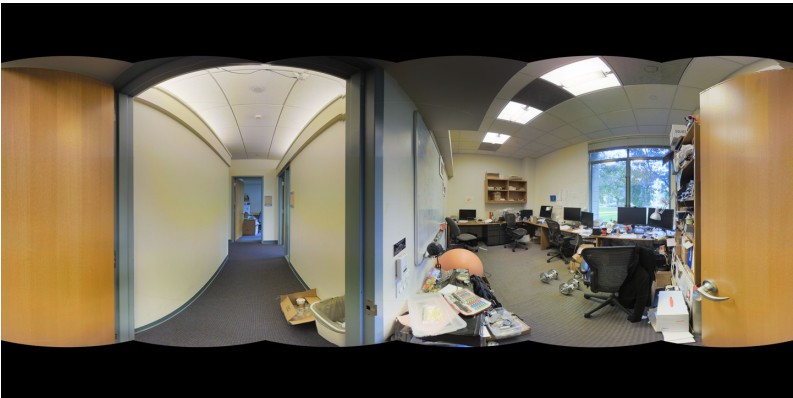

**Example 1 Description:**
Consider the impact of the equirectangular projection (ERP) on the lower polar area. The floor's carpet pattern curves outward due to ERP distortion, appearing unnaturally stretched. Despite this warping, the carpet's neutral color and repeating design remain distinguishable, with their consistency helping to visually anchor the space. When analyzing furniture placement, the semicircular arrangement of tables and chairs is an illusion–distortion causes three rectangular tables (placed in three directions) to appear curved. Zooming in reveals black-backed chairs around the tables, with papers, staplers, notebooks, and other office supplies scattered on the tables; the actual layout is angular, not rounded. Moving upward from the floor, bookshelves line the walls, filled with books and small decorative items like plants or figurines. Framed certificates and photographs hang nearby, suggesting a workspace for academics or professionals. These details contrast with the distortion but remain sharp enough to convey the space's purpose.

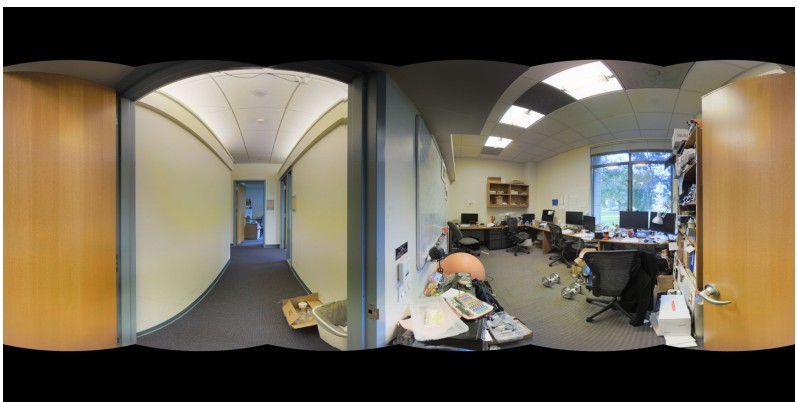

**Example 2 Description:**
The image uses an equirectangular projection (ERP), which stretches and distorts objects near the polar regions. In ERP, the top edge represents the "north pole" of the 360° scene, meaning objects near the top may appear warped compared to their real-world arrangement. In the top left polar area, artificial lights are positioned at the junction of the white ceiling and two walls. Due to ERP's distortion of vertical lines, these lights likely form part of a corner where two walls meet the ceiling in reality. A small wooden board hanging on the side suggests this area is a distinct workspace separated by a wall from the right side. The top right polar area features square recessed ceiling lights and a blackboard with writing; the wall separation implies a different functional zone. Despite distortion, the square recessed lights suggest they are aligned in a straight line on the actual ceiling, while the blackboard's vertical placement on the wall is evident. The left and right areas are divided by a wall but exist in the same polar region–distortion may make them appear adjacent in the projection, but functionally, they are distinct workspaces. The ceiling lights and boards on both sides serve their respective zones while maintaining proximity for collaboration.

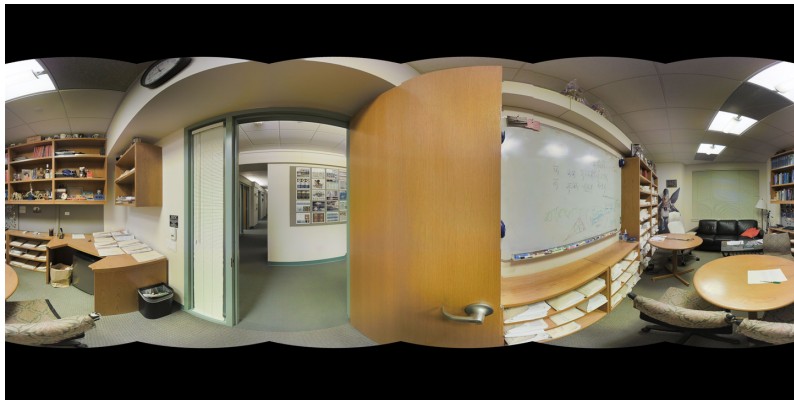

**Example 3 Description:**
The scene uses a 360-degree equirectangular projection (ERP), which distorts polar-region objects, causing straight objects like doors to appear curved and the ceiling to look warped. A circular clock near the ceiling remains unobstructed, but ERP stretching complicates perceptions of occlusion. A large door on the right (visually curved due to ERP) partially obscures a whiteboard; their positional relationship persists–if the door is closer to the viewer, it logically blocks the whiteboard behind it. A hanging shelf near the whiteboard also partially blocks items on it. ERP affects visual representation but not physical reality: the door and shelf are positioned to occlude other objects, and their distortion does not erase their spatial relationships. The partial visibility of the whiteboard and shelf items confirms that occlusion occurs despite ERP warping.

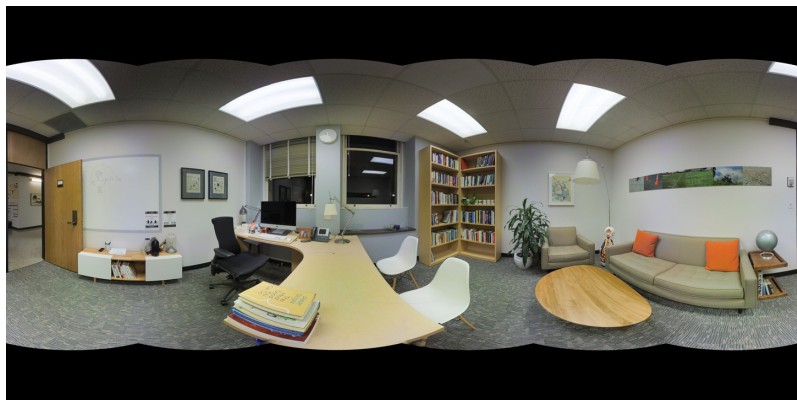

**Example 4 Description:**
This is a 360° equirectangular projection (ERP), which distorts polar regions, but does this mean objects do not interact? In the top polar region, artificial lights are "embedded" in the ceiling grid, implying connection or overlap with the grid structure–a clear interaction. In the bottom polar region, books are stacked on a rectangular tabletop (direct physical contact), and an oval coffee table "touches" the carpeted floor. Despite projection-induced distortion of the coffee table's legs, the contact point (table-to-floor) remains, confirming interaction persists.

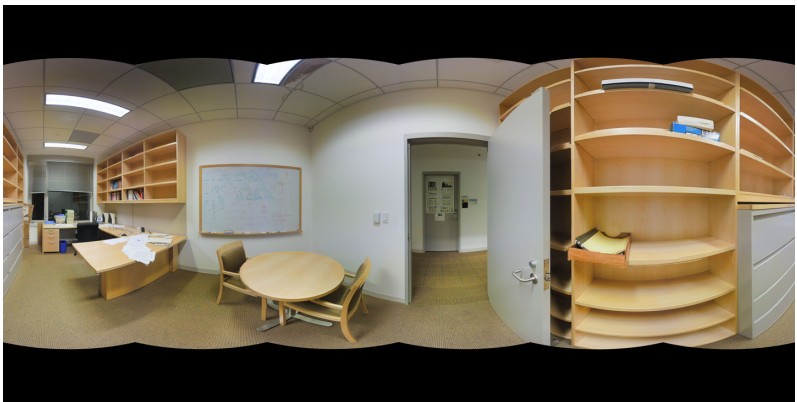

**Example 5 Description:**
The image uses an equirectangular projection, which distorts polar regions, making straight objects near the top appear curved or stretched in the projection (though they are physically straight). The upper polar region shows parts of the ceiling, where curvature from distortion affects fluorescent lights (appearing bent), flat ceiling panels in a grid, air vents, and minor ceiling discolorations following the warped pattern. On the right side, a bookshelf extends into this region; its shelves appear slightly warped in the projection but are actually level. The top shelf holds items like a soundbar, books, and storage boxes, whose edges look curved here but are rectangular in real space.

---

# B  OMNIVQA DATASET DETAILS

This section provides a detailed breakdown of the question types, templates, and representative examples that constitute the OmniVQA dataset. The questions are designed to evaluate a model's capabilities in object recognition in polar regions, attribute analysis, and spatial reasoning within panoramic images. Table 7 lists these categories along with their distribution percentages.

Table 7: Question types, templates, and examples for the OmniVQA dataset.

| Question Type | Question Template | Question Example |
|---|---|---|
| **(i) Object Identification (1,510 questions, 31.12%)** | | |
| Object Identification: Recognizing objects within polar regions of panoramic images. | What object is presented in the [polar region]? | What object is presented in the top polar region of the image? |
| | Which objects can be identified in the [polar region]? | Which objects can be identified in the bottom polar area? |
| **(ii) Object Attribute Analysis (1,255 questions, 25.87%)** | | |
| Object Attribute Analysis: Describing visual attributes such as characteristics, shape etc. | What shape does the object in the [polar region] exhibit? | What shape does the object in the top polar region exhibit? |
| | What visual features can you observe about the object in the [polar region]? | What visual features can you observe about the object in the upper polar area? |
| **(iii) Spatial Relationship Reasoning (2,087 questions, 43.01%)** | | |
| Spatial Relationship Reasoning: Inferring spatial relationships among multiple objects. | What is the spatial relationship between [object A in the polar region] and [object B near it]? | What is the spatial relationship between the object in the polar region and the object near it? |
| | What is the spatial relationship between objects in the [polar region]? | What is the spatial relationship between objects in the top polar regions? |

## C  TRAINING VISUALIZATION

### REWARD AND COMPLETION STATISTICS DURING RL TRAINING

This section presents the learning curves for key reward signals and completion statistics over the course of RL post-training. Each plot corresponds to a major reward or output property, providing insight into how the model adapts and improves throughout training.

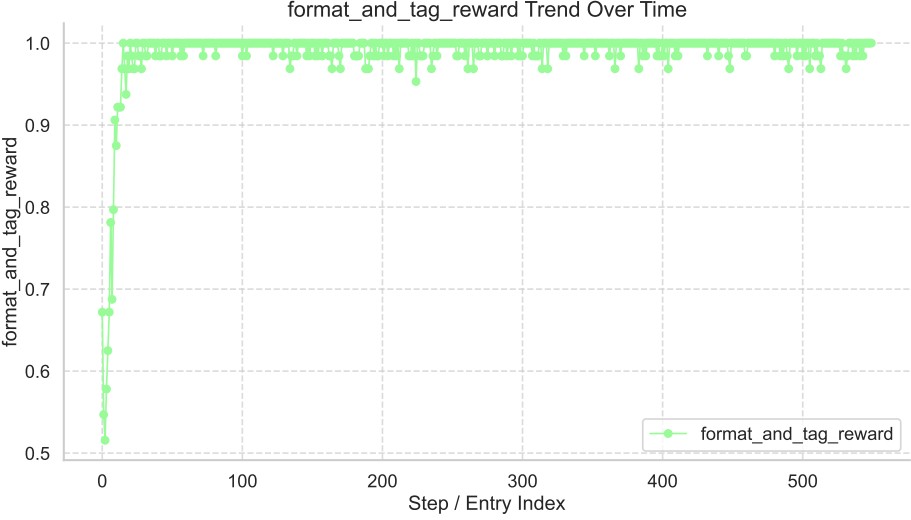

Figure 5: Format and Tag Reward Across Training Steps.

**Interpretation:** Figure 5 tracks the *format_and_tag_reward* as a function of training steps (0–550). Initially, the reward hovers near 0.50, indicating that early model outputs only partially comply with the desired formatting and tagging standards. During the first 200 steps, the reward rises quickly above 0.75, then continues a steady upward trend, reaching close to 0.98 by step 500. This near-linear increase demonstrates that the model rapidly learns and maintains strong adherence to prescribed output formatting.

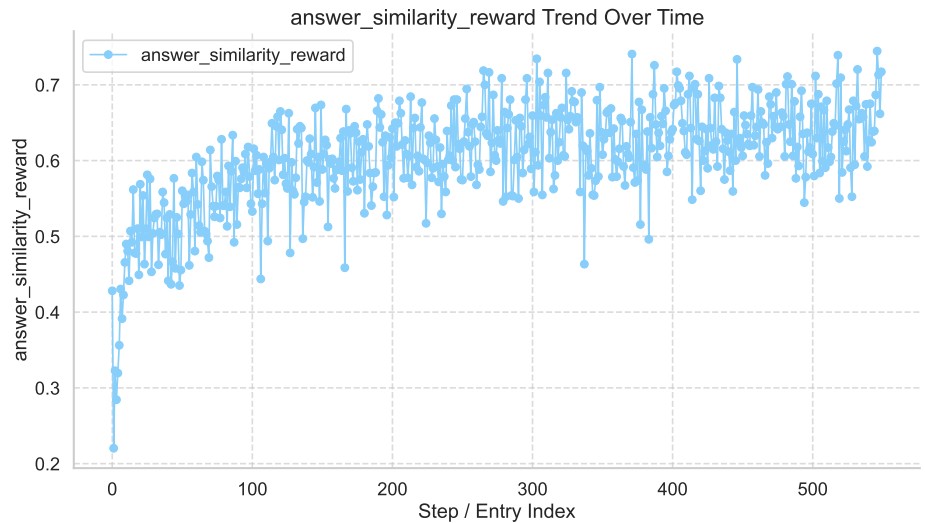

Figure 6: Answer Similarity Reward Over Training Steps.

**Interpretation:** Figure 6 illustrates the progression of the *answer_similarity_reward* during training. The curve starts at approximately 0.25, reflecting a substantial gap between initial model answers and reference responses. Over the first 100–150 steps, the reward increases sharply to about 0.45, then climbs more gradually to around 0.65 by step 500. This consistent upward trend indicates that the model's generated answers become increasingly semantically and lexically aligned with the ground-truth references as training advances.

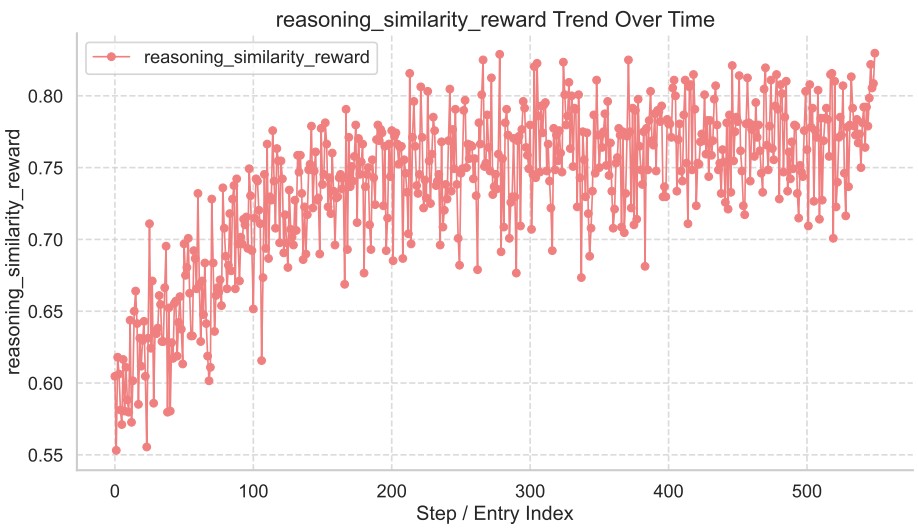

Figure 7: Reasoning Similarity Reward Over Training Steps.

**Interpretation:** Figure 7 presents the trajectory of the *reasoning_similarity_reward*. Starting at around 0.55—indicating modest alignment between generated and reference reasoning chains—the reward rises most rapidly within the first 150 steps, then continues to grow steadily, approaching 0.80 by step 500. This pattern demonstrates effective learning of structured, expert-like reasoning, with the final high reward signifying close conformity to the reference chains of thought.

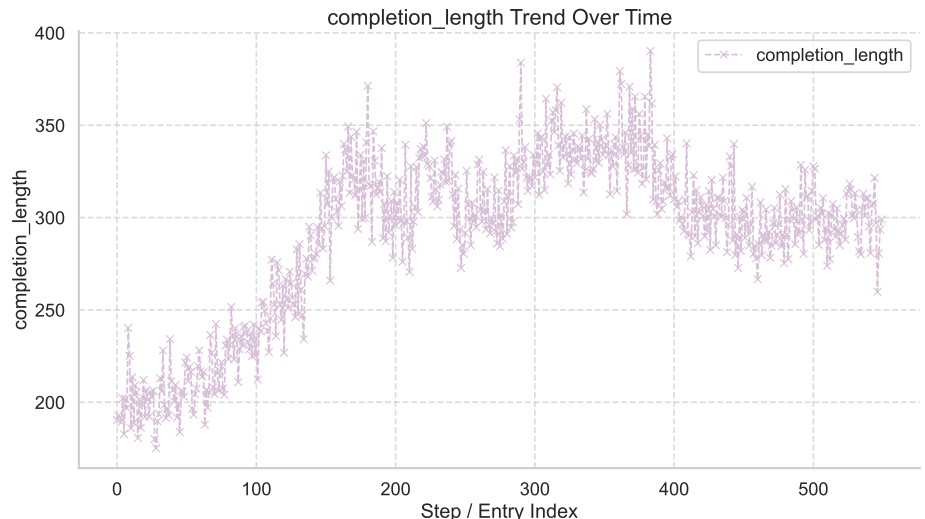

Figure 8: Generated Completion Length Over Training Steps.

**Interpretation:** Figure 8 shows the trend in generated completion length (measured in tokens or characters) over training. Early outputs average around 250 units, corresponding to relatively brief responses. Between steps 100 and 300, there is a marked increase to roughly 325, as the model produces more detailed and elaborate outputs. After step 300, length stabilizes in the 350–380 range, reflecting a balance between completeness and conciseness. This upward shift highlights the model's enhanced capacity to provide comprehensive, well-explained answers.

# D LARGE LANGUAGE MODELS USAGE STATEMENT

We disclose our use of Large Language Models (LLMs). An LLM was indeed used only for language polishing (e.g., grammar, spelling, clarity, tone) on text whose content and structure were created by the authors. No substantive changes to claims, data interpretation, or conclusions were introduced by the LM.

Beyond language polishing, LLMs played a significant role in our research methodology between December 2024 and August 2025. The authors take full responsibility for all content, including LLM-generated outputs, which were manually verified to ensure accuracy. For our dataset creation, we utilized `GPT-4o` and `DeepSeek-R1` via their public APIs to generate visual descriptions and Chain-of-Thought (CoT) reasoning. Concurrently, the `Qwen2.5-VL` and `Qwen2.5-14B` models, downloaded directly from Hugging Face, were used for description generation and answer summarization.

Furthermore, `DeepSeek-V3` was employed via its public API for two critical functions in our pipeline: first, as the automated judge for our DeepSeekScore evaluation metric, and second, as the reward model providing semantic similarity scores for our GRPO training framework. The specific prompts governing all LLM-driven tasks are detailed in Appendix A.

