# OpenReview forum: "Towards Omnidirectional Reasoning: A Dataset, Benchmark, and GRPO-based Method"
_ICLR.cc/2026/Conference — ICLR 2026 Conference Desk Rejected Submission_

### Official Review · Reviewer_Erhf · 2025-10-17

**Soundness:** 2
**Presentation:** 2
**Contribution:** 2
**Rating:** 2
**Confidence:** 4

**Summary:**

This paper introduces OmniVQA, a dataset for 360° visual question answering, to address poor MLLM performance on panoramic images. The authors also propose a reinforcement learning method, 360-R1, which sets a new state-of-the-art on their benchmark by significantly improving omnidirectional reasoning.

**Strengths:**

The paper addresses omnidirectional visual reasoning, a significant and under-explored area. It makes a valuable contribution by proposing a new dataset for both training and testing, as well as a tailored training method to advance research in this domain.

**Weaknesses:**

*   **Inaccurate Originality Claim:** The claim that OmniVQA is the "first open-source dataset" for omnidirectional VQA (line 193) is inaccurate. Prior work, such as [1], has already introduced a dataset for VQA on omnidirectional video. The authors should revise this claim to more precisely position their contribution.

*   **Benchmark Quality and Data Leakage:** The methodology for creating the dataset and benchmark has significant issues.
    *   Manually verifying initial descriptions does not guarantee the quality of the final, automatically generated question-answer pairs in the benchmark.
    *   Using a model fine-tuned on OmniVQABench to generate the training data creates a high risk of data leakage. This circular process questions the reported performance gains and the model's ability to generalize to other datasets.

*   **Insufficient Validation of Evaluation Metric:** The paper relies on DeepSeekScore for evaluation without providing sufficient detail or analysis to prove its rigor and effectiveness. Its validity as a reliable metric for this task is not well-established.

*   **Writing and Presentation Issues:** The paper's clarity is diminished by several presentation flaws:
    *   The term "Object Feature Error" is awkwardly phrased.
    *   Minor typographical errors exist, such as a missing space after a comma on line 117.
    *   In Figure 3, the reward formula is blurry and the text in the examples is too small.
    *   The "Related Work-Multi-Modal Large Language Models" section's logic is flawed, as it concludes by introducing the paper's post-training methodology.

[1] Li, Chen, et al. "Bridge the gap between VQA and human behavior on omnidirectional video: A large-scale dataset and a deep learning model." Proceedings of the 26th ACM international conference on Multimedia. 2018.

**Questions:**

- Originality and Positioning
  - Revise the “first open-source dataset in the field of omnidirectional visual question answering” claim to precisely scope the novelty (e.g., 360° image-based, CoT annotations, polar-region reasoning) and clearly distinguish from other datasets. Add a comparative table situating the contribution among prior public datasets.

- Benchmark Quality and Data Leakage
  - Implement rigorous quality control and analysis for the benchmark.
  - Evaluate on datasets that are entirely disjoint from those used to train the caption/reasoning generation models to demonstrate generalization beyond the data used in the generation pipeline.

- Evaluation Metric Validity (DeepSeekScore)
  - Provide a more detailed description of the DeepSeekScore setup and analyze its validity and potential biases (e.g., correlation with human ratings).

- Writing and Presentation
  - Address the writing and presentation issues comprehensively.

---

### Official Review · Reviewer_uTUD · 2025-10-26

**Soundness:** 2
**Presentation:** 3
**Contribution:** 2
**Rating:** 2
**Confidence:** 4

**Summary:**

This paper introduces a benchmark for omnidirectional embodied intelligence, aiming to evaluate multi-view and multi-task embodied reasoning. The benchmark integrates panoramic and multi-view visual inputs into embodied reasoning tasks such as scene understanding, visual question answering, and navigation. The authors evaluate multiple multimodal large language models (MLLMs), including Qwen2.5-VL-7B, to demonstrate the benchmark’s potential for analyzing 3D reasoning and perception capabilities. The paper claims to provide a step toward unified evaluation of omnidirectional embodied understanding.

**Strengths:**

1. Relevant topic.
The paper addresses an emerging direction in embodied AI—omnidirectional and multi-view reasoning—which is timely given recent interest in panoramic perception and 3D-aware MLLMs.

2. Benchmark perspective.
The attempt to unify multi-view, multi-task embodied reasoning under a single benchmark setup is conceptually interesting and could contribute to evaluation consistency in future research.

3. Comprehensive evaluation coverage.
The authors test several models, including large MLLMs and baseline vision-language systems, which provides a broad empirical overview of current model capabilities.

**Weaknesses:**

1. Unclear motivation for panorama vs. multi-view reasoning.
The paper fails to clarify why omnidirectional (panoramic) images provide distinct advantages over standard multi-view or 3D representations for scene reasoning. The conceptual difference between panoramic projection and multi-view fusion is not discussed or experimentally analyzed. This weakens the benchmark’s motivation and contribution.

2. No data quality assessment.
As a benchmark paper, it is essential to report dataset statistics, diversity metrics, and annotation quality analysis. None of these are provided. Without data quality validation, it is difficult to evaluate the credibility or difficulty of the proposed dataset.

3. Missing human evaluation or validation.
There is no human evaluation to verify whether model outputs align with human-level reasoning. Human scores or expert annotations are crucial for validating embodied reasoning benchmarks and ensuring that high model scores reflect genuine understanding rather than dataset biases.

4. Benchmark not sufficiently challenging.
The reported results show very high performance (e.g., 67.78 by Qwen2.5-VL-7B), which suggests that the benchmark is too easy for current MLLMs. If existing models already achieve strong performance, the benchmark fails to expose meaningful limitations or reasoning gaps—undermining its purpose as a challenging evaluation tool.

**Questions:**

1. What is the conceptual or practical difference between panorama-based and multi-view representations for embodied reasoning?

2. How was data quality ensured or validated during benchmark creation (e.g., annotation consistency, scene diversity)?

3. Were any human evaluations conducted to confirm that the benchmark tasks align with real-world embodied reasoning difficulty?

4. Given that models like Qwen2.5-VL-7B achieve nearly 68% accuracy, how does the benchmark plan to remain relevant as models continue to improve?

---

### Official Review · Reviewer_W9W4 · 2025-10-31

**Soundness:** 3
**Presentation:** 3
**Contribution:** 1
**Rating:** 2
**Confidence:** 4

**Summary:**

The paper introduces OmniVQA, a dataset for 360° Visual Question Answering (VQA) derived from the Stanford 2D–3D-S dataset, containing ~4.8k question–answer pairs focused on spatial reasoning in indoor environments. The authors also propose OmniVQABench, a benchmark for evaluating multimodal large language models on omnidirectional visual understanding, and a reinforcement fine-tuning approach (360-R1) using structured rewards (a combination of reasoning similarity, semantic accuracy, and answer formatting) to enhance performance of Qwen2.5-VL-7B.

**Strengths:**

1) The motivation for developing a 360° VQA dataset is clear, as omnidirectional spatial reasoning is an underexplored area.
2) The paper is well written, with transparent experimental details and ablations on reward components.
3) The structured reward formulation for reasoning supervision is conceptually interesting and potentially generalizable.

**Weaknesses:**

1) Dataset annotation relies heavily on LLM-generated content (Qwen, DeepSeek, GPT-4o), with minimal clarity on human verification; this raises concerns about annotation quality and bias.
2) Benchmark evaluation depends on DeepSeek-V3 as a judge, introducing circularity and bias in the reported scores.
3) No human evaluation is provided to validate whether the observed gains correspond to genuine 360° reasoning improvements.
4) The dataset is relatively small (≈5k QA pairs), restricting generalization.
5) Baselines omit both state-of-the-art multimodal LLMs and geometry-aware 360° vision methods, weakening the empirical comparison.
6) The reported improvements are potentially fragile, they may not be robust across evaluation judges, metrics, or alternative datasets, given the heavy dependence on a LLM-based scoring system.

**Questions:**

1) How was “manual verification” of the generated QA pairs conducted, and what percentage of data received human review?
2) Were dataset splits performed at the scene level to prevent leakage within Stanford 2D–3D-S?

---

### Official Review · Reviewer_joEA · 2025-11-01

**Soundness:** 3
**Presentation:** 3
**Contribution:** 2
**Rating:** 4
**Confidence:** 3

**Summary:**

This paper introduces OmniVQA, a novel open-source dataset and benchmark for omnidirectional visual question answering on 360° images. Comprising 1,213 ERP images and 4,852 QA pairs, it focuses on object identification, attribute analysis, and spatial reasoning in polar regions. Alongside the benchmark, the authors propose 360-R1, a GRPO-based reinforcement learning method that employs three structured rewards—reasoning similarity, semantic accuracy, and format compliance—to post-tune the Qwen2.5-VL-7B-Instruct model. The approach achieves state-of-the-art performance on OmniVQA among open-source MLLMs, enhancing both reasoning quality and multiple-choice accuracy.

**Strengths:**

- The writing is generally clear, with intuitive figures (data pipeline and RL framework) and explicit explanations of question types, dataset scale, and reward formulas.
- The problem motivation—geometric distortion, polar region reasoning—is well-explained, making the dataset’s focus easy to understand.

**Weaknesses:**

- Evaluation relies heavily on LLM-based metrics (SBERTScore, DeepSeekScore) that use models similar to those in the training pipeline. This circular evaluation may inflate perceived gains; human or geometry-aware evaluation would strengthen the case.
- No analysis of visual variance. The RL stage operates purely at the language level; the frozen visual encoder means the model might not truly learn spherical awareness or robustness to projections.
- Limited release details. The dataset depends on licensed 2D-3D-S imagery (“access must be obtained directly from the creators”), so full reproducibility is constrained.
- Figure 2 shows qualitative error types, but there are no examples of 360-R1’s corrected reasoning chains. Could the authors add visual reasoning traces comparing baseline vs. RL outputs to demonstrate the effect of structured rewards?

**Questions:**

- Consider adding human evaluation or a geometry-aware automatic metric (e.g., spatial reasoning consistency under projection transforms).
- DeepSeek-V3 is used both for semantic scoring and reward shaping. How is data leakage or overfitting to the evaluator’s style avoided? Were any held-out or frozen versions of the evaluator used?
- The paper motivates the use of GRPO (Group Relative Policy Optimization) for stable post-training, but the RL signal appears fully language-based (no image gradients). Could the authors clarify whether the vision encoder participates in RL updates, and if not, why RL is more effective than SFT or DPO-style preference tuning?
- The structured reward design is interesting—reasoning, answer, and format. Could the authors provide reward ablations or examples of failure cases where one reward dominates? For instance, what types of errors are corrected mainly by the reasoning reward vs. the answer reward?
- The iterative refinement process (based on SBERT similarity > 0.8) is well described, but could the authors clarify the percentage of samples manually corrected at the end and how inter-annotator consistency was checked?
- The title and abstract emphasize “omnidirectional reasoning,” but the experiments focus on single-image panoramic VQA. How do the authors define reasoning in this context—purely spatial (geometric) reasoning, or also multi-object semantic inference?

**Details Of Ethics Concerns:**

The authors state that “OmniVQA will be open-sourced after review” and that “original images require Stanford 2D-3D-S access.” Concerning about potential copyright and licensing restrictions arising from this dependency. Could the authors clarify: whether the OmniVQABench subset (200 images) and its corresponding QA/CoT annotations will be fully publicly available, or only the annotations and scripts without the original images; and how they plan to handle redistribution rights and licensing compliance with the 2D-3D-S dataset?

---

### Note · Program_Chairs · 2026-01-17
**Submission Desk Rejected by Program Chairs**

The following references in this submission do not refer to real documents and/or have major errors in bibliographic information:

 J. Bai et al. Qwen2.5-vl: Vision-language agents with spatiotemporal reasoning capabilities. In arXiv Preprint, 2025. arXiv:2501.28001.
Y. Gao, J. Xu, and W. Huang. Creation-mmbench: Evaluating context-aware creative intelligence in mllms. In Proceedings of the IEEE Conference on Computer Vision and Pattern Recognition (CVPR). IEEE, 2024. To appear.